# From Despair to Hope: First Arabic Experience of ^177^Lu-PSMA and ^161^Tb-PSMA Therapy for Metastatic Castration-Resistant Prostate Cancer

**DOI:** 10.3390/cancers16111974

**Published:** 2024-05-23

**Authors:** Akram Al-Ibraheem, Ahmed Saad Abdlkadir, Deya’ Aldeen Sweedat, Stephan Maus, Ula Al-Rasheed, Samer Salah, Fadi Khriesh, Diyaa Juaidi, Dina Abu Dayek, Feras Istatieh, Farah Anwar, Aisha Asrawi, Alaa Abufara, Mohammad Al-Rwashdeh, Ramiz Abu-Hijlih, Baha’ Sharaf, Rami Ghanem, Hikmat Abdel-Razeq, Asem Mansour

**Affiliations:** 1Department of Nuclear Medicine, King Hussein Cancer Center (KHCC), Al-Jubeiha, Amman 11941, Jordan; aa.15389@khcc.jo (A.S.A.); ds.14021@khcc.jo (D.A.S.); ua.13155@khcc.jo (U.A.-R.); djuaidi@khcc.jo (D.J.); da.13175@khcc.jo (D.A.D.); fi.15108@khcc.jo (F.I.); 2School of Medicine, University of Jordan, Al-Jubeiha, Amman 11942, Jordan; 3Department of Nuclear Medicine, Saarland University Medical Center, D-66421 Homburg, Germany; stephan.maus@uks.eu; 4Department of Medicine, King Hussein Cancer Center (KHCC), Amman 11941, Jordan; shsalah@moh.gov.sa (S.S.); aa.10579@khcc.jo (A.A.); mal-rwashdeh@khcc.jo (M.A.-R.); bs.13628@khcc.jo (B.S.); habdelrazeq@khcc.jo (H.A.-R.); 5Department of Nuclear Medicine, Klinikum Fulda, Pacelliallee 4, 36039 Fulda, Germany; fadi.khreish@klinikum-fulda.de; 6Department of Nuclear Medicine, Warith International Cancer Institute, Karbala 56001, Iraq; f.anwar@warith-ici.net; 7Department of Nursing, King Hussein Cancer Center (KHCC), Amman 11941, Jordan; aa.16111@khcc.jo; 8Department of Radiation Oncology, King Hussein Cancer Center (KHCC), Amman 11941, Jordan; rhijlih@khcc.jo; 9Department of Surgery, King Hussein Cancer Center (KHCC), Amman 11941, Jordan; rghanem@khcc.jo; 10Department of Diagnostic Radiology, King Hussein Cancer Center (KHCC), Amman 11941, Jordan; amansour@khcc.jo

**Keywords:** mCRPC, prostate cancer, radioligand therapy, specific membrane antigen, ^177^Lu-PSMA, ^161^Tb-PSMA, PSMA therapy, PSMA-RLT, PSA, PRLT

## Abstract

**Simple Summary:**

The recent approval of [^177^Lu]Lu-prostate-specific membrane antigen (PSMA) for managing metastatic castration-resistant prostate cancer (mCRPC) has catalyzed the innovation of various PSMA-targeted radiopharmaceuticals. In our retrospective study, we explored both safety and efficacy of two beta-emitting PSMA radioligands, [^177^Lu]Lu and [^161^Tb]Tb, for mCRPC therapy. Our study included 53 patients and reinforced prior evidence validating the clinical safety and efficacy of these radioligands. Our research suggested that these treatments are characterized by a favorable safety profile with negligible toxicity. Moreover, [^161^Tb]Tb-PSMA recipients, though trialed in a smaller patient sample, yielded concordant outcomes on par with those receiving [^177^Lu]Lu-PSMA, highlighting its promise as an alternative therapy and warrants additional investigation.

**Abstract:**

The objective of this retrospective study is to assess the effectiveness and safety of two beta-emitting prostate-specific membrane antigen (PSMA) radioligands, [^177^Lu]Lu and [161Tb]Tb, in heavily treated patients with metastatic castration-resistant prostate cancer (mCRPC). A total of 148 cycles of beta-emitting PSMA radioligand therapy were given to 53 patients at a specialized cancer care center in Amman, Jordan. This treatment was offered following the exhaustion of all prior treatment modalities. Approximately half of the cases (n = 26) demonstrated an initial partial response to PSMA radioligand therapy. Moreover, roughly one-fourth of the patients (n = 13) exhibited a sustained satisfactory biochemical response, which qualified them to receive a total of six PSMA radioligand therapy cycles and maintain continued follow-up for additional treatment cycles. This was reflected by an adequate prostate-specific antigen (PSA) decline and a concomitant partial response evident on [^68^Ga]Ga-PSMA positron emission tomography/computed tomography imaging. A minority of patients (n= 18; 34%) experienced side effects. Generally, these were low-grade and self-limiting toxicities. This study endorses previous research evidence about PSMA radioligand therapy’s safety and efficacy. It also provides the first clinical insight from patients of Arab ethnicity. This should facilitate and promote further evidence, both regionally and internationally.

## 1. Introduction

The incidence of prostate cancer (PC) in the Arab world is expected to experience an increase, despite currently having a lower age-standardized rate compared to Europe [1]. It is believed that the true incidence of PC among the Arab population is underestimated. This fact stems from a significant number of patients being diagnosed at advanced stages, coupled with underdiagnosis of early-stage disease due to inadequate screening rates and limited public awareness, rather than an accurate reflection of the disease’s prevalence. [2,3,4]. Ethnic variation and tumor aggressiveness have also been proposed as factors contributing to this divergent incidence pattern [2,3,4,5]. Recent studies have shown a continued rise in new PC cases in Jordan [6], where it is currently the fifth most common cancer among males nationally and the second most common globally [6,7].

Metastatic castration-resistant PC (mCRPC), a form of advanced PC, is an ongoing and serious challenge [8]. Despite the recent increase in treatment options, it remains an invariably fatal condition, demonstrating resistance to the majority of current treatment modalities [8]. Prostate-specific membrane antigen (PSMA) is found to be significantly upregulated in mCRPC. PSMA is highly overexpressed in more than 90% of mCRPC lesions [9]. When patients with mCRPC experience disease progression, they are considered for radioligand therapy (RLT) as a treatment option after exhausting other conventional hormonal and chemotherapy approaches [9]. Numerous studies have shown that PSMA RLT effectively improves biochemical control, improves quality of life, and alleviates pain [10,11,12,13,14,15,16,17,18,19,20,21,22]. Furthermore, the current body of evidence indicates that PSMA RLT leads to higher progression-free survival (PFS) and overall survival (OS) in mCRPC patients [10,23,24,25,26,27]. The observed low toxicity further supports the safety of [^177^Lu]Lu-PSMA RLT in this patient population.

The labeling of PSMA with the beta-particle-emitting radionuclide [^177^Lu]Lu has been approved for the treatment of mCRPC by regulatory agencies in the United States and Europe since 2022 [28]. Nevertheless, this agent was incorporated into the theranostic institutional practice at King Hussein Cancer Center (KHCC) in Amman, Jordan, in 2017, making it one of the pioneering institutions in the Arab World to adopt this targeted treatment [28,29,30]. In addition to its therapeutic benefits, treatment with beta-emitting [^177^Lu]Lu facilitates post-therapeutic scintigraphic imaging to assess the localization of the radiotracer [31]. This theranostic approach can be complemented with a positron-emitting [^68^Ga]Ga-PSMA diagnostic counterpart for diagnostic and monitoring purposes [32]. Another beta radiotracer, namely the [^161^Tb]Tb-PSMA radioligand, possesses physical and biological distribution comparable to [^177^Lu]Lu-PSMA (Table 1), which has been recently used at KHCC (Figure 1) [33,34].

As the use of PSMA radioligand therapy continues to expand through promising clinical trials [35], we conducted this study to provide evidence from real-world experience with beta-emitting PSMA therapeutic agents in mCRPC patients of Jordanian and Arab descent.

## 2. Materials and Methods

### 2.1. Institutional Review Board Approvals

This retrospective study obtained ethical approval from the Research Ethics Board at the King Hussein Cancer Center in Amman, Jordan. The research project was granted approval under the reference number (21 KHCC 43) before its implementation.

### 2.2. Patient Selection

An institutional cancer registry of PC patients was recovered from the tertiary cancer care center in Amman, Jordan. In this registry, a total of 556 patients were identified with PC from 2017 up until the study conception date. The number of medical referrals for each of these patients was obtained and examined. Out of the total number of patients, only 68 patients were referred to the nuclear medicine department to determine their eligibility for PSMA RLT. These patients were referred from the genitourinary oncology department following the exhaustion of all available therapeutic options as determined by a multidisciplinary team. Among these, only 61 were found eligible for PSMA RLT. Patients who had a less than two PSMA RLT cycles were excluded from the study. Ultimately, a total of 53 patients were considered eligible for inclusion in the study (Figure 2).

### 2.3. Data Collection

Demographic, clinical, biochemical, histopathological, radiological, and detailed medication histories were obtained for each of the 53 patients included in the study. This information included the patient’s age at diagnosis, baseline Eastern Cooperative Oncology Group (ECOG) performance status, nationality, date of PC diagnosis, and follow-up duration. The primary PC histopathologic examination data from baseline were analyzed to determine the type of prostate cancer and Gleason score. Baseline radiology reports from CT scans were reviewed to assess the metastatic pattern and areas of involvement in the nodal and distant metastatic domains. Additionally, information about prior therapeutic modalities received before PSMA RLT was collected, including the type and duration of past therapies. The study collected serial biochemical profile datasets for each patient, including parameters such as complete blood profile (CBP), liver function test (LFT), renal function test (RFT), PSA, and alkaline phosphatase (ALP). Data specific to PSMA RLT were analyzed, including serial trends for PSA, CBC, LFT, RFT, ALP, and pain scoring before and after therapy. Molecular imaging conducted around the relevant timepoint of PSMA therapy, such as [^18^F]Fluorodeoxyglucose ([^18^F]FDG) PET/CT and/or [^68^Ga]Ga-PSMA PET/CT, was also reviewed to provide baseline and follow-up information. Additionally, any reported symptomatic side effects and subjective changes in quality of life from the patient’s medical records were collected. Data organization was performed using Microsoft Excel Professional Plus 2021.

### 2.4. PSMA RLT Procedure

No-carrier-added [^177^Lu]LuCl_3_ and [^161^Tb]TbCl_3_ were radiolabeled to PSMA-617 for therapeutic use. For [^177^Lu]Lu-PSMA-617, quality control was conducted using thin-layer chromatography, while for [^161^Tb]Tb-PSMA-617, high-performance liquid chromatography was utilized, ensuring a radiochemical purity exceeding 95%. Patients were administered [^177^Lu]Lu-PSMA-617 in doses ranging from 6.0 to 8.0 GBq per treatment cycle, and [^161^Tb]Tb-PSMA-617 at 5.5 GBq per cycle. Both treatments were given intravenously with supplemental oral hydration to aid in excretion and minimize radiation to non-target tissues. The treatment intervals were set at a minimum of 7 weeks apart for [^177^Lu]Lu-PSMA-617 and 8 weeks apart for [^161^Tb]Tb-PSMA-617.

### 2.5. Response Assessment

We analyzed initial biochemical PSA response using the criteria established by the PC Working Group 3 [36]. A PSA response represented at least a 50% decrease from the start of treatment, while progression was considered when an increase of 25% or more exceeding 2 ng/mL was witnessed, confirmed by a second measurement 3 weeks later, with values falling between these limits categorized as stable. We also assessed the best overall PSA response, which was determined by considering the minimum PSA levels attained within a four-week period following the completion of PSMA RLT cycle. A significant ALP response was defined as a total decline of 30% [36]. [^68^Ga]Ga-PSMA PET/CT imaging was assessed and was usually conducted after 2–3 cycles of RLT and subsequently every 2 cycles thereafter. [^68^Ga]Ga-PSMA PET/CT imaging response was assessed according to a recent consensus [37]. Partial response was characterized by a reduction in uptake or tumor volume of more than 30%; stable disease by up to 30% change in uptake or tumor volume without new lesions; and progressive disease by the appearance of two new lesions, >30% increase in uptake or tumor volume, or the expansion of diffuse bone marrow involvement. [^18^F]FDG PET/CT were performed for some patients. This was assessed to determine and ensure no discrepancy with [^68^Ga]Ga-PSMA PET/CT to exclude tumor heterogeneity or dedifferentiation [38,39].

### 2.6. Toxicity Profile

The study involved serial assessments of hematological parameters (hemoglobin, leukocytes, and platelets), key biomarkers (PSA and ALP), LFT, and RFT (via estimated creatinine levels). These evaluations were collected before initiating each PSMA RLT cycle, 2 to 4 weeks after each PSMA RLT cycle, and at 6- to 12-week intervals during the ongoing follow-up. In addition, a comprehensive examination of the patients’ medical records was performed pre- and post-PSMA RLT cycles to identify any therapy-induced symptomatic complaints, with an emphasis on detailing these occurrences. The classification of the severity of any adverse events was conducted in alignment with the Common Terminology Criteria for Adverse Events, version 5.0 [40].

### 2.7. Statistical Analysis

The statistical analysis was conducted using Stata software version 17 (Stata Corporation, College Station, TX, USA). The normality of variables was assessed using the Shapiro–Wilk W test. Normally distributed values were reported as mean ± standard deviation (SD) with range, while values that did not follow a normal distribution were reported as median and interquartile range (IQR). Categorical values were presented as frequencies and proportions. Changes in PSA and ALP levels in individual patients, categorized by the magnitude of change, were visually represented using waterfall plots.

## 3. Results

### 3.1. Baseline Profiles

This retrospective study included a cohort of 53 consecutive patients diagnosed with mCRPC. All these patients were diagnosed with acinar prostatic adenocarcinoma. The mean age at diagnosis was 64 years, with a SD of ±9.2 years. The following information outlines the baseline characteristics of the study participants (Table 2).

### 3.2. Patients’ Nationality

The analysis of the demographic data collected indicates that the study encompasses experiences from both national and regional perspectives. The majority of the participants enrolled in this study are of Jordanian descent. Furthermore, a significant portion of the study sample, approximately 20%, comprises individuals from non-Jordanian Arab populations, as detailed in Table 3.

### 3.3. Pre-PSMA RLT Plan

Past medical and surgical history examination reveals a comprehensive regimen of systemic treatments, encompassing hormonal therapy, chemotherapy, and radiotherapy. Notably, the patients underwent extensive pretreatment, receiving a median of five different therapies. These treatments were administered over a median period of 38 months, with an IQR between 25 to 70 months. The hormonal therapies included abiraterone, enzalutamide, and bicalutamide. Additionally, taxane-based chemotherapy agents such as docetaxel, paclitaxel, and cabazitaxel were utilized. Surgical interventions, such as radical prostatectomy or metastasectomy, were performed on a small subset of the patients included in the study. Furthermore, some patients received radiotherapy targeting either the primary tumor or metastatic sites (Table 4).

### 3.4. PSMA RLT

In this retrospective study, a total of 148 cycles of beta-emitting PSMA RLT were administered, with the majority being [^177^Lu]Lu-PSMA cycles (n = 144). The median number of [^177^Lu]Lu-PSMA cycles provided to patients was four, with an interquartile range (IQR) of 3 to 5 cycles. The median treatment activity for [^177^Lu]Lu-PSMA RLT was 22.2 GBq, with an IQR of 14.8 to 29.6 GBq. In contrast, the mean administered radioactivity for the [^161^Tb]Tb-PSMA treatments was 11.1 GBq, with a SD of ±3.8 GBq. All patients underwent at least two PSMA RLT cycles, which were administered at a median interval of 2 months, with an IQR of 2 to 3 months.

### 3.5. Efficacy

#### 3.5.1. PSA Response According to Prostate Cancer Clinical Trials Working Group

Following the initial cycle of beta-emitting PSMA RLT, 26 out of the 53 treated patients (i.e., 49% of the whole studied cohort) exhibited a partial biochemical response, as evidenced by a ≥50% decline in PSA levels (Figure 3). 

Subsequently, eight of these patients managed to maintain a stable PSA profile over the next three cycles, and half of this subgroup (four patients) continued to exhibit stability for a total of six cycles. Additionally, nine patients demonstrated a cumulative partial response after receiving four cycles, with two of them sustaining this response through six cycles. The remaining seven patients sustained a partial biochemical PSA response for the administered six PSMA RLT cycles. Conversely, 6 out of the total 26 initial PSA responders experienced biochemical progression and were transitioned to palliative care. Notably, the remaining three patients are currently under observation and are expected to undergo further PSMA RLT cycles in the near future (Figure 4).

Otherwise, 14 out of the 53 treated patients achieved PSA stability, and 13 more patients experienced PSA progression following four PSMA RLT cycles. All of them were deemed unresponsive due to evident disease progression on PSMA PET imaging or discordant FDG PET/CT imaging mandating a shift to palliative care. A more detailed set of radionuclide-based results are shared below (Table 5). 

It is worth noting that all patients received [^177^Lu]Lu-PSMA RLT initially and that [^161^Tb]Tb-PSMA was offered for the four non-responding bimodal benefactors (Figure 4). A single cycle of [^161^Tb]Tb-PSMA RLT was administered to each of these patients, with three continuing follow-up care and one transitioning to palliative care.

#### 3.5.2. Best PSA Response

For [^177^Lu]Lu-PSMA receivers, the lowest post-therapeutic PSA was achieved after the median of one cycle (IQR of 1–2), and its median value was 29.6 (IQR of 7.7–67.3) ng/mL. In total, 29 out of the 53 individuals who received [^177^Lu]Lu-PSMA treatment exhibited a partial PSA response. Additionally, only five patients experienced a progressive increase in PSA levels, whereas all others demonstrated stable PSA trends (Figure 5).

#### 3.5.3. ALP Biochemical Response

Only 34 mCRPC patients exhibited abnormal ALP levels before the initiation of PSMA RLT. Subsequent examinations were conducted on this subgroup in order to assess alterations in ALP patterns subsequent to PSMA RLT administration. After the first cycle of [^177^Lu]Lu-PSMA RLT, a total of 15 patients experienced a significant reduction in ALP levels, achieving a decline of 30% or more (Figure 6). 

Among the 26 patients who were identified as partial responders based on PSA levels, 6 also demonstrated a concurrent decline of 30% or more in ALP levels. Additionally, during the ongoing administration of [^177^Lu]Lu-PSMA RLT, two more patients were observed to achieve a reduction of 30% or more in ALP levels. With regard to [^161^Tb]Tb-PSMA RLT, only one patient demonstrated a significant ALP reduction (Table 6).

#### 3.5.4. [^68^Ga]Ga-PSMA PET/CT Response Evaluation

Initial imaging response assessments conducted using [^68^Ga]Ga-PSMA PET/CT, typically 2–3 cycles after the commencement of PSMA RLT, indicated a partial response in 27 patients, stable disease in 16 patients, and disease progression in 10 patients. Among those who initially responded to the therapy based on PSA levels, 22 patients showed a partial response on [^68^Ga]Ga-PSMA PET/CT imaging, while the remainder exhibited stable disease.

### 3.6. Safety

Only a minority experienced significant adverse effects following any cycle of beta-emitting PSMA RLT. Clinically, gastrointestinal symptoms, specifically nausea and vomiting, were the most commonly reported adverse effects, affecting 18 patients. From a biochemical perspective, hematotoxicity emerged as the most prevalent adverse effect. Notably, hematotoxicity also represented the most severe form of toxicity, with four patients experiencing serious conditions. Among these, one patient developed both thrombocytopenia and anemia concurrently, necessitating a blood transfusion for effective management, while the remaining three patients suffered from single cell-lineage hematotoxicity, managed conservatively. Additionally, a single case of grade-4 nephrotoxicity was encountered. This particular case required urgent medical intervention, including fluid replacement and a nephrology consultation. Apart from these instances, all other reported toxicities were self-limiting and reversible (Table 7).

## 4. Discussion

This retrospective study confirmed the therapeutic efficacy of beta-emitting PSMA RLTs, namely [^177^Lu]Lu and [^161^Tb]Tb, in managing mCRPC patients. Pioneering the Arab region, our study offers robust real-world evidence supporting the clinical safety and effectiveness of these agents. Our findings demonstrate an initial significant positive response in the majority of patients, as shown by substantial decreases in PSA and ALP levels, as well as improvements in imaging results. Additionally, our study highlights the favorable safety profile of PSMA RLT, with manageable side effects observed in a small percentage of patients, further confirming the safety and effectiveness of this nuclear medicine therapy. This paper presents the initial clinical experience of PSMA RLT in the Arabic context, aiming to highlight its potential and raise awareness regarding the significance of adopting this innovative technique in both the fields of nuclear medicine and oncologic theranostics [41].

For metastatic PC, the primary treatment option is hormonal therapy [42]. This involves removing androgens through the use of androgen deprivation therapy, which typically can prevent disease progression for up to 18–24 months [42]. However, if the disease progresses despite this therapy, mCRPC can develop [43]. The use of theranostic agents such as [^177^Lu]Lu-PSMA is an emerging therapy for mCRPC that is available in many countries worldwide [44,45]. In parallel, [^68^Ga]Ga-PSMA imaging can provide high diagnostic accuracy for disease staging, monitoring response, restaging, and assessing eligibility for PSMA RLT [46,47,48]. 

Recently, there are numerous clinical trials that have been conducted to evaluate the efficacy of [^177^Lu]Lu-PSMA in the treatment of advanced mCRPC [14,49,50,51]. In a phase II trial conducted by Tagawa et al., 47 patients were treated with [^177^Lu]Lu-PSMA, and 55.3% of those patients had also received prior chemotherapy [50]. The results showed that 10.6%, 36.2%, and 59.6% of patients, respectively, had a ≥50%, ≥30%, or any decrease in PSA after a single dose of 177Lu-PSMA therapy [50]. In a similar previous attempt, Hofman et al. recruited 30 patients with mCRPC who had previously used standard therapy options [49]. Patients received 1 to 4 cycles of intravenous [^177^Lu]Lu-PSMA at six-week intervals [49]. Disease progression status and adverse reactions were assessed. In addition to the primary study endpoint of the PSA response rate (a 50% decrease from baseline), other endpoints, such as PFS, OS, imaging responses and quality of life (QOL), were also evaluated [49]. Briefly, 17 (57%) patients met the PSA response criteria, and 29 (97%) of the patients had some degree of PSA response [49]. Moreover, 3 months after the last injection, [^68^Ga]Ga-PSMA imaging demonstrated that 10%, 53%, and 30% exhibited nonprogressive disease, encompassing complete response, partial response, and stable disease, respectively [49]. The authors also reported a PSA regression in 27 (90%) patients, and the median PFS and OS were 7.6 and 13.5 months, respectively [49]. The study also revealed that [^177^Lu]Lu-PSMA therapy was well tolerated with minimal adverse events, such as dry mouth, in 87% of the patients [49]. Moreover, the pain level of the study participants (27 patients, or 90%) improved at all study time points, which significantly contributed to quality of life [49]. More recently, Yadav et al. investigated the efficacy of [^177^Lu]Lu-PSMA therapy in a cohort of 90 patients with mCRPC [14]. They found that after the first cycle of therapy, 32.2% of patients experienced a greater than 50% decline in PSA levels and that this percentage increased to 45.5% by the end of the study (after up to seven cycles) [14]. Additionally, 27.5% of patients achieved partial remission, 43.5% had stable disease, and 29% had progressive disease [14]. Furthermore, 54% of patients reported an improvement in pain score after the first cycle of therapy [14]. In 2018, Kim et al. conducted a meta-analysis and gathered available data from 10 retrospective series including 455 patients who reported favorable outcomes [52]. It revealed that approximately 2/3 of any decrease in PSA and 1/3 of any decrease in PSA greater than 50% can be expected after the first cycle of [^177^Lu]Lu-PSMA therapy [52]. Subsequently, in a recent systematic review conducted by Patell et al., the authors performed a comprehensive qualitative analysis of the existing literature [53]. The authors determined that [^177^Lu]Lu-PSMA had shown a minimal toxicity profile and was generally well tolerated in male patients with advanced metastatic disease that is unresponsive to conventional treatments [53]. Both retrospective analyses and prospective clinical trials have indicated that PSMA RLT is a viable treatment option for individuals with mCRPC who have not responded to traditional hormonal therapies and chemotherapy [53]. Table 8 summarizes the key similarities and difference between our study results and results of above discussed studies in this paragraph.

PSMA ligand therapy delivers radiation to non-target tissues that express PSMA. However, the side effects of this therapy are generally mild and related to the absorbed dose [18,54,55]. The most exposed organs are the kidneys, salivary glands, lacrimal glands, and bone marrow. However, hematotoxicity is the most common serious adverse event, occurring in 12% of patients with a high tumor burden in bone [54]. Xerostomia is currently the second most common side effect reported [14,54]. This adverse effect is most frequently encountered in patients who do not perform any protective measures, e.g., salivary gland cooling [49]. Regarding the toxicity profile, [^177^Lu]Lu-PSMA is generally recognized for its high safety standards, with only minimal self-limiting toxicities reported thus far. A recent meta-analysis reviewed all prior studies to affirm the safety of [^177^Lu]Lu-PSMA. This analysis included 23 clinical investigations, all of which aligned with the specified research objectives. Notably, the majority of published studies reported only low-grade toxicities. Hemoglobin-related toxicities were the most prevalent, affecting 23% of patients. Additionally, severe thrombocytopenia occurred in 15% of patients, while leukopenia was noted in 14.2% of patients. Nephrotoxicity was observed in 9.5% of patients, and 14.5% reported salivary gland toxicities, including pain, swelling, and dry mouth. Other less common toxicities included fatigue, nausea, and loss of appetite.

Currently, PSMA RLT is used as a last resort in the treatment of mCRPC. However, it is unclear whether this therapy would be more effective if it was used in first-line or earlier settings. Comparative studies are needed to investigate this issue. Combination strategies are also emerging as interesting alternative options. Both the STAMPEDE and CHAARTED studies revealed that combination therapies can be effective in treating mCRPC [56,57]. Basu et al. reported that adding [^177^Lu]Lu-PSMA to abiraterone therapy can be effective at mitigating side effects [58]. Crumbaker et al. reported that [^177^Lu]Lu-PSMA plus idronoxil is a safe and effective treatment option [59]. Enzalutamide has been demonstrated to increase the expression of PSMA, a key factor in the effectiveness of PSMA RLT [60]. Research has shown that treatment with enzalutamide can lead to a significant rise in PSMA uptake in cancer cells, potentially enhancing the therapeutic outcomes of subsequent Lu-PSMA treatments [61]. Clinical studies have observed increased levels of PSMA uptake following treatment with enzalutamide, indicating a potential for improved treatment efficacy. Additionally, combining enzalutamide with Lu-PSMA has shown promising results in terms of prostate-specific antigen progression-free survival, suggesting a synergistic effect between the two treatments [62]. This combination therapy may offer benefits in terms of overall survival and disease progression, potentially making it a valuable approach in the management of prostate cancer. Overall, [^177^Lu]Lu-PSMA shows great promise as part of a combination therapy.

The exploration of [^161^Tb]Tb-PSMA RLT in mCRPC is a promising avenue of investigation, with ongoing clinical trials focused on elucidating its safety and efficacy profile [35]. Early findings suggest that the [^161^Tb]Tb-PSMA may offer a favorable therapeutic index and demonstrate potential efficacy in terms of PSA decline and tumor burden reduction. However, a more comprehensive understanding of its safety and efficacy in mCRPC patients awaits further data from ongoing trials. Preliminary dosimetry assessments revealed that [^161^Tb]Tb-PSMA had a slightly higher absorbed doses to organs at risk than [^177^Lu]Lu-PSMA RLT but notably higher doses to tumor lesions, suggesting a potentially superior therapeutic index [63]. A recent case study illustrated significant partial remission and PSA decline in an advanced mCRPC patient post-[^161^Tb]Tb-PSMA RLT, suggesting therapeutic benefits even in patients who progressed following extensive [^177^Lu]Lu-PSMA RLT [64]. The efficacy of [^161^Tb]Tb-PSMA is under investigation, with a focus on PSA response rates, radiological PFS, OS, and quality of life [35]. Ongoing trials such as the VIOLET study and the REALITY trial in Germany aim to evaluate the efficacy and safety of [^161^Tb]Tb-PSMA RLT in mCRPC patients, with anticipated results to provide further insights into its clinical utility [35]. These trials represent significant endeavors in advancing the understanding and potential application of [^161^Tb]Tb-PSMA RLT in the management of mCRPC, contributing to the evolving landscape of PC therapeutics.

Our research aligns with the existing literature on the subject, demonstrating the initial effectiveness of [^177^Lu]Lu-PSMA RLT while also confirming its safety. Additionally, we provide a comprehensive analysis of the safety and efficacy of [^161^Tb]Tb-PSMA, a novel treatment option [10,11,12,13,14,15,16,17,18,19,20,21,22]. Despite the inherent limitations of a single-center, retrospective design, and a small cohort size, our investigation stands as a pioneering contribution, offering the first documented real-world experience of RLT in metastatic castration-resistant prostate cancer (mCRPC) patients on both national and regional scales.

## 5. Conclusions

Our findings confirm the therapeutic potential of beta-emitting PSMA radioligands [^177^Lu]Lu and [^161^Tb]Tb in mCRPC management, with minimal toxicity and notable biochemical response. Such observed evidence from a real-world experience aims to reinforce the potential of PSMA RLT as a viable treatment option, meriting further exploration both nationally and regionally.

## Figures and Tables

**Figure 1 cancers-16-01974-f001:**
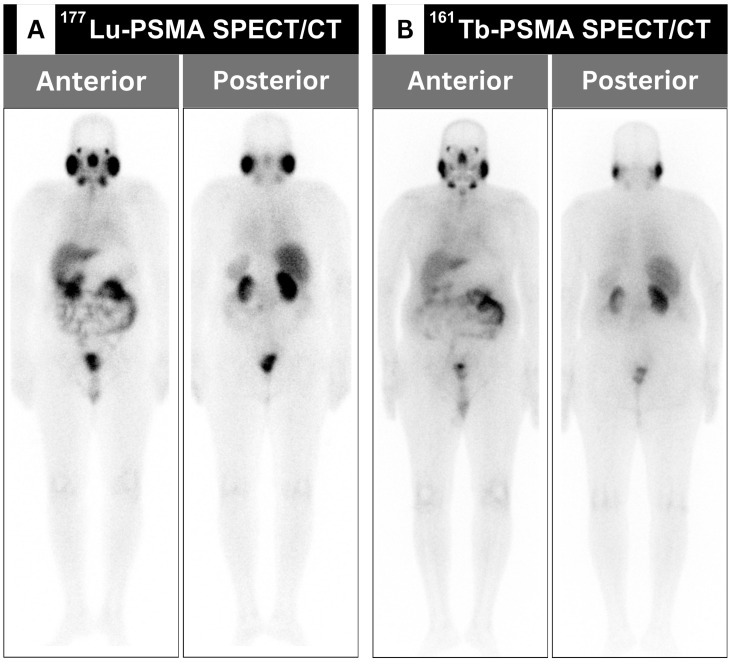
A group of normal anterior and posterior planar scintigraphic images of a single patient for (**A**) [^177^Lu]Lu-prostate-specific membrane antigen (PSMA) and (**B**) [^161^Tb]Tb-PSMA therapies. Notably, there is an unremarkable difference in the physiologic distribution of both therapeutic agents within the lacrimal glands, salivary glands, hepatobiliary, genitourinary, and gastrointestinal systems.

**Figure 2 cancers-16-01974-f002:**
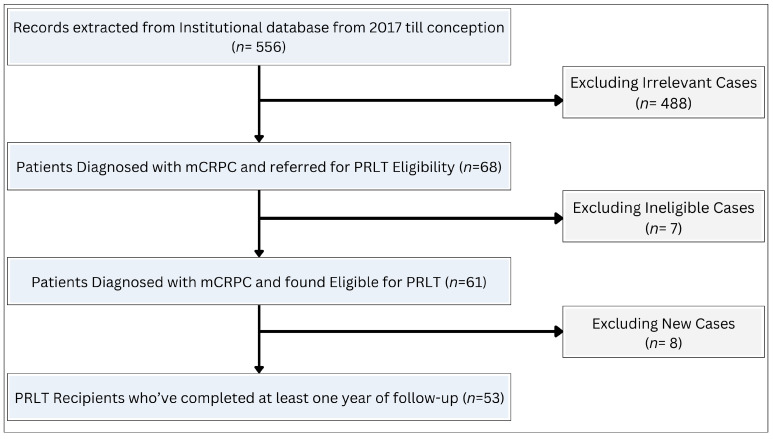
Flowchart illustrating the process of patient selection.

**Figure 3 cancers-16-01974-f003:**
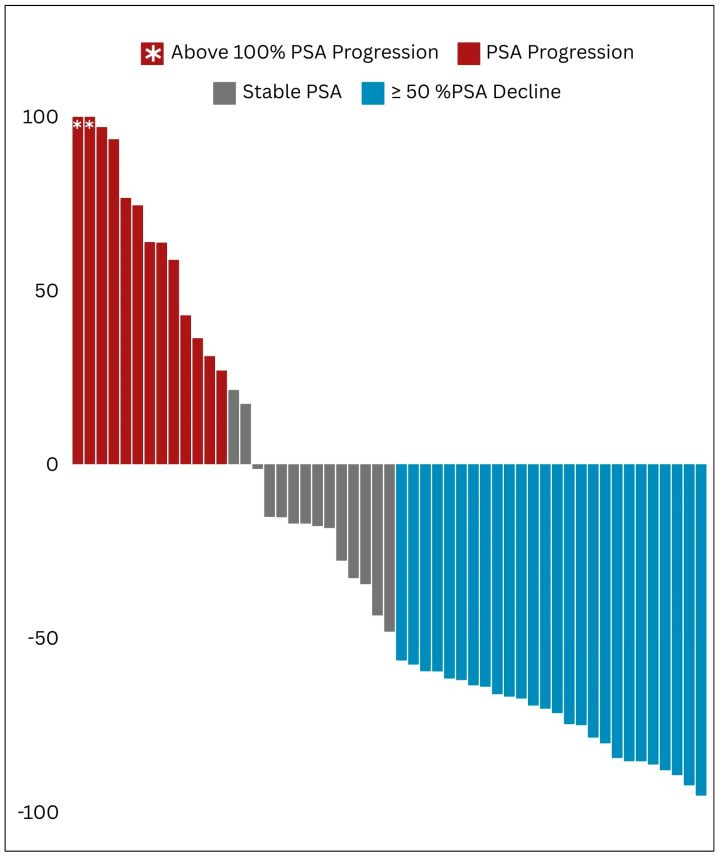
Waterfall plot illustrating the initial patterns of response in prostate-specific antigen (PSA) levels.

**Figure 4 cancers-16-01974-f004:**
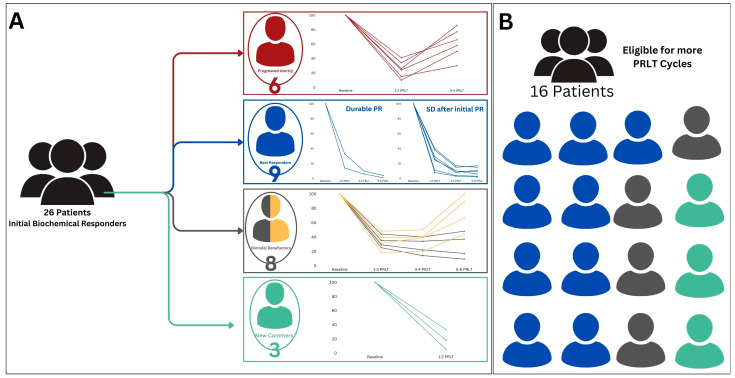
(**A**) Schematic summary for patients exhibiting initial PSA response as depicted by PSA percentile trending plots. (**B**) Graphical representation of the total number of patients who qualified for further [^177^Lu]]Lu-PSMA RLT cycles.

**Figure 5 cancers-16-01974-f005:**
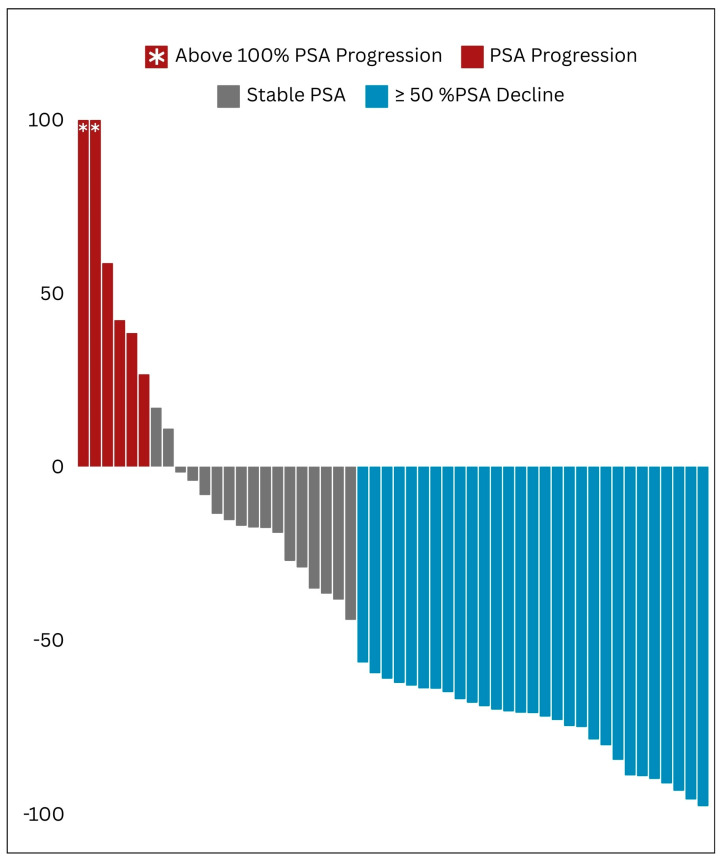
Waterfall plot illustrating the best biochemical response achieved in prostate-specific antigen (PSA) trends.

**Figure 6 cancers-16-01974-f006:**
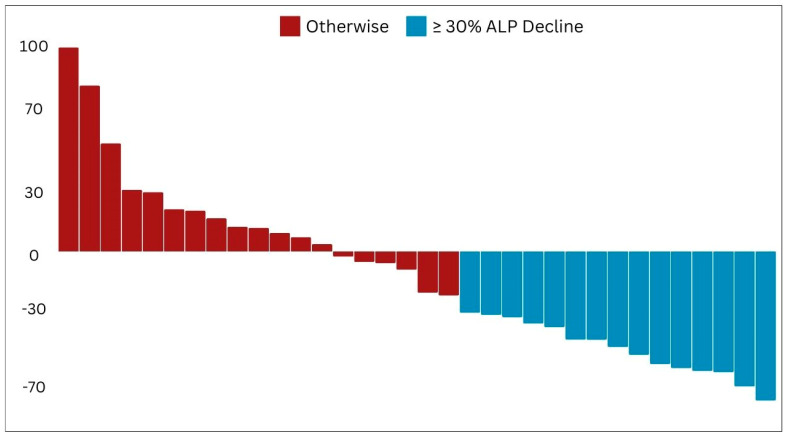
Waterfall plot illustrating the initial patterns of response in alkaline phosphatase levels.

**Table 1 cancers-16-01974-t001:** A summary of the key similarities and differences between the two agents.

Parameter	[^177^Lu]Lu-PSMA ^1^	[^161^Tb]Tb-PSMA
Abundance of Auger and conversion electrons	Lower abundance of Auger and conversion electrons	Higher abundance of Auger and conversion electrons
Physical Half-Life	Approximately 6.7 days	Approximately 6.9 days
Active non-target system	Urinary system	Urinary system
Energy for Beta Emission	Mean kinetic energy of 134 keV	Mean energy of 154 keV
Energy for Gamma Emission	Gamma rays emitted at energies of 113 keV (6.23%) and 208 keV (10.4%)	Gamma rays emitted at lower energies of 48.9 keV (17.0%) and 74.6 keV (10.2%)
Cellular Absorbed Dose	Effective in delivering dose to tumor cells, but potentially less efficient than [^161^Tb]Tb-PSMA	More than 3-fold-higher cellular absorbed dose when compared to [^177^Lu]Lu-PSMA
Current Status	Approved for the treatment of metastatic castration-resistant prostate cancer	Under current research in clinical trials and investigations
Empiric Dose estimate	Up to 7.4 GBq	Up to 5.5 GBq

^1^ PSMA: prostate-specific membrane antigen.

**Table 2 cancers-16-01974-t002:** Baseline characteristics of the study cohort.

**Age at Diagnosis of mCRPC ^1^ (in Years)**	
Mean	64 years
Standard Deviation	9.2 years
**Initial Gleason Score**	**(Number, Percentage)**
7	7, 13.2%
8	15, 28.3%
9	28, 52.8%
10	3, 5.7%
**Initial ECOG ^2^ Performance Status**	**(Number, Percentage)**
0	23, 43.4%
1	30, 56.6%
**Bone Pain**	**(Number, Percentage)**
Present	42, 79.2%
Absent	11, 20.8%
**Metastatic Sites**	**(Number, Percentage)**
Bone	48, 90.5%
Lymph nodes	39, 73.6%
Lung	2, 3.8%
Liver	2, 3.8%
Adrenal	2, 3.8%
**Biochemical Profile**	**(Median, IQR ^3^)**
PSA (ng/mL)	49.4, 9.5–117
Leukocyte (10^3^/µL)	7.5, 6.2–9.3
Creatinine (mg/dL)	0.9, 0.7–1.1
Alkaline Phosphatase (U/L)	103, 79.5–164
**Hematological Profile**	**(Mean ± SD ^4^)**
Hemoglobin (g/dL)	13.2 ± 1.7
Platelet Count(10^3^/µL)	245 ± 70.6

^1^ mCRPC: metastatic castration-resistant prostate cancer; ^2^ ECOG: Eastern Cooperative Oncology Group (ECOG) performance status. ^3^ IQR: interquartile range; ^4^ SD: standard deviation.

**Table 3 cancers-16-01974-t003:** Original nationality of the enrolled patients.

Nationality	(Number, Percentage)
Jordanian	42, 79.2%
Palestinian	7, 13.2%
Iraqi	2, 3.8%
Syrian	1, 1.9%
Sudanese	1, 1.9%

**Table 4 cancers-16-01974-t004:** Past therapeutic history.

**Surgical Intervention**	**(Number, Percentage)**
Bilateral Orchidectomy	6, 11.3%
Prostatectomy	12, 22.6%
Metastasectomy	4, 7.5%
**Androgen Deprivation Hormonal Therapy**	**(Number, Percentage)**
Goserelin	40, 75.5%
Abiraterone	39, 73.5%
Enzalutamide	27, 54.7%
Bicalutamide	23, 43.4%
Decapeptyl	10, 18.9%
**Chemotherapy**	**(Number, Percentage)**
Docetaxel	38, 71.7%
Paclitaxel	1, 1.9%
Cabazitaxel	1, 1.9%
**Other Therapies**	**(Number, Percentage)**
Radiotherapy	35, 66%
Ketoconazole Medication	2, 3.8%

**Table 5 cancers-16-01974-t005:** Radionuclide-based results for the administered PSMA RLT.

**[^177^Lu]Lu-PSMA ^1^**		
**Entity**	**Number**	
Total cycles	144	
Patients Received	53	
Patients with Initial Partial PSA ^2^ Response	25	
Patients with Initial PSA stability	12	
Patients with Initial PSA uprise	12	
**[^161^Tb]Tb-PSMA**		
**Entity**	**Results**	**Details**
Total cycles	4	5.5 GBq each
Patients Received	4	After [^177^Lu]Lu-PSMA failure
Patients with Initial Partial PSA Response	1	69.8% decline in PSA
Patients with Initial PSA stability	2	PSA decline of 16.7%, and 29.5%
Patients with Initial PSA uprise	1	PSA uprise of 71.3%

^1^ PSMA: prostate-specific membrane antigen; ^2^ PSA: prostate-specific antigen.

**Table 6 cancers-16-01974-t006:** Detailed ALP trends for [^161^Tb]Tb-PSMA receivers.

[^161^Tb]Tb-PSMA ^1^	
Patients	ALP ^2^ trend
Patient 1	Decline of 43.6%
Patient 2	Uprise of 13.8%
Patient 3	Uprise of 28.1%
Patient 4	Uprise of 4.4%

^1^ PSMA: prostate-specific membrane antigen; ^2^ ALP: alkaline phosphatase.

**Table 7 cancers-16-01974-t007:** Observed side effects following beta-emitting PSMA RLT.

**[^177^Lu]Lu-PSMA ^1^**					
**Variable**	**Any Grade**	**Grade 1**	**Grade 2**	**Grade 3**	**Grade 4**
Nausea	18	16	2		
Dry Mouth	10	10			
Fatigue	15	12	3		
Anemia	5	3	1	1	
Thrombocytopenia	15	6	7	2	
Leukopenia	13	7	4	2	
↑^2^ LFT ^3^	8	6	2		
↑ Creatinine	10	9			1
**[^161^Tb]Tb-PSMA**					
**Variable**	**Any Grade**	**Grade 1**	**Grade 2**	**Grade 3**	**Grade 4**
Fatigue	2	1	1		
↑ Creatinine	1	1			

^1^ PSMA: prostate-specific membrane antigen; ^2^ ↑: Increased; ^3^ LFT: liver function test.

**Table 8 cancers-16-01974-t008:** Selected earlier research works examining the efficacy and/or safety of [^177^Lu]Lu-PSMA in advanced prostate cancer, with emphasis on significant similarities and differences from our cohort.

Study Name ^1^	Study Type	Primary Aim	Key Similarities	Key Differences
Yadav et al., 2020 [14]	P-S ^2^	S ^3^ and E ^4,^	An initial decline of more than 50% in PSA^5^ was recorded in up to 46% of patients, which closely resembles our initial biochemical response rate. [^177^Lu]Lu-PSMA ^6^ safety was confirmed with self-limiting gastrointestinal toxicity most frequently observed.	The study also examined the impact of [^177^Lu]Lu-PSMA on survival.
Kim 2018 [52]	M-A ^7^	E	Any decline in PSA trends was achieved in 68% of patients, which closely resembles our obtained response rate of about 71%.	An initial decline of more than 50% in PSA was recorded in only one third of the patients; however, this was a pooled analysis of an old data (back in 2018), which included a predominance of preliminary studies.
Patell 2023 [53]	S-R ^8^	S and E	The study affirmed the safety and efficacy of [177Lu]Lu-PSMA. This was observed in a systematic review of 100 previously published papers on the topic of interest.	Unlike our study, a qualitative rather than quantitative approach was implemented.
Hofman 2018	C-T ^9^	S and E	An initial decline of more than 50% in PSA was recorded in 57% of patients, endorsing [177Lu]Lu-PSMA efficacy. In our cohort, it was 49%. Safety was affirmed.	Unlike our study, xerostomia was the most prevalent side effect, followed by gastrointestinal self-limiting complaints. A predominant number of patients (nearly 90%) in this clinical trial were previously exposed to taxane-based chemotherapy, which might have affected the toxicity profile.
Groener 2021	R-S ^10^	S and E	An initial decline of more than 50% in PSA was recorded in up to 46% of patients, which closely resembles our biochemical responder rate (49%). An initial significant decline in ALP ^11^ was witnessed in 22% of patients. In our cohort, 15 out of 34 who initially presented with elevated ALP experienced a more than 30% decline.	All the patients developed hematotoxicity of any grade, while low-grade nephrotoxicity was observed in 75% of the patients. The inclusion of patients with previous critical illnesses (like baseline bone marrow suppression and chronic kidney disease) might have affected the evaluation of the safety profile.

^1^ Study name: derived from combining the name of first author with the year of publication; ^2^ P-S: prospective study; ^3^ S: safety; ^4^ E: efficacy; ^5^ PSA: prostate-specific antigen; ^6^ PSMA: prostate-specific membrane antigen; ^7^ M-A: meta-analysis; ^8^ S-R: systematic review; ^9^ C-T: clinical trial; ^10^ R-S: retrospective study; ^11^ ALP: alkaline phosphatase.

## Data Availability

The data presented in this study are available on request from the corresponding author. The data are not publicly available owing to privacy.

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
