# Peer review of "From Despair to Hope: First Arabic Experience of 177Lu-PSMA and 161Tb-PSMA Therapy for Metastatic Castration-Resistant Prostate Cancer"

_cancers, 2024, doi:10.3390/cancers16111974_

Round 1

Reviewer 1 Report

Comments and Suggestions for Authors

In this clinical research work, the authors conducted a retrospective analysis of the efficacy and safety of two PSMA-targeted radiopharmaceuticals Lu177 and Tb161. This work will be a representative reference for mCRPC treatment.  Minor issues needed to be addressed:

1. Physics comparison between Lu177 and Tb161 will be helpful in understanding the treatment pipelines of these two radiopharmaceuticals.

2. As mentioned in the discussion section, in PSMA based treatment, most exposed non-targeted organs are kidneys, salivary glands, lacrimal glands, and bone marrow. The toxicity will depend on the pharmacokinetics of  radiopharmaceuticals in these organs, more relevant discussions on the PSMA non-targeted toxicity are needed.

Author Response

Dear Reviewer 1
Thank you very much for your insightful comments. Kindly note that we have chosen to subdivide each question with number labeling to facilitate the review process and answer each question accordingly.

Below are the answers to your respectful review points.

  1. Physics comparison between Lu177 and Tb161 will be helpful in understanding the treatment pipelines of these two radiopharmaceuticals.
  • Thank you for this impactful suggestion. This was added and discussed in (Table 1; yellow highlights).

  1. As mentioned in the discussion section, in PSMA based treatment, most exposed non-targeted organs are kidneys, salivary glands, lacrimal glands, and bone marrow. The toxicity will depend on the pharmacokinetics of radiopharmaceuticals in these organs, more relevant discussions on the PSMA non-targeted toxicity are needed.  
  • Thank you for this impactful suggestion. We have detailed toxicity patterns reported in most recent meta-analysis in lines 384-393 (yellow highlights) to provide more insightful details.

Reviewer 2 Report

Comments and Suggestions for Authors

The manuscript submitted by Al-Ibraheem reports responses from radioligand therapy of using 177Lu-PSMA and 161-Tb-PSMA to treat castration-resistant prostate cancer. 177Lu-PSMA has been approved by the US FDA two years ago, and lots of clinical data have been reported. Nevertheless, this is the first report of a clinical trial from Arabian population, and could be used to compare with the results reported by others and obtained from different ethnic groups. Listed below are some suggested changes:

·       There are some grammatical errors. For example, “We also assessment the best overall PSA response …” (Line 158), “Surgical interventions, including radical prostatectomy or metastectomy.” (Line 215, not a complete sentence), and “To date, there are a several clinical trials …” (Line 328).

·       Table 1: Initial ECOG Performance Status for 1 should be 43.4%, not 44.4%.

·       Table 1, footnote: “3SD” should be “4SD” and there should be a “4” at “SD” of the “Mean ± SD”.

·       Section 3.4: “The median treatment activity for [177Lu]Lu-PSMA RLT was 22.2 GBq, with an IQR of 14.8 to 29.6 GBq. In contrast, the mean administered radioactivity for the [161Tb]Tb-PSMA treatments was 11.1 GBq, with a SD of ± 3.8 GBq.”. Why did the authors use two different expression methods (median and IQR vs mean and SD) for the dosages of [177Lu]Lu-PSMA and [161Tb]Tb-PSMA? Ideally, they should be the same, so it would be easier to compare their differences.

·       Table 6: “18” for Any Grade and “16” for Grade 1. What are the grade for the other 2 patients?

·       Lines 340-342: “Moreover, 3 months after the last injection, 40%, 37%, and 37% of patients had nonprogressive disease (complete response, partial response, and stable disease), respectively”. Double check the numbers as 40% + 37% + 37% is > 100%.

·       The Discussion section is poorly written. This section is used to discuss the reported data and compare them with those reported by others. However, for the entire Discussion section, the authors mentioned only the data reported by others, and there were no comparisons at all with their own data. For example, in the 3rd paragraph, the authors mentioned the response rates reported by others. The authors should compare their response rates with those reported by others, and see if they are consistent. If not, the authors should provide rationales for the inconsistency. Similarly, for the 4th and 5th paragraphs, the authors mentioned the prediction factors reported by others. The authors should compare their data with those reported by others and see if they are consistent. The authors should provide rationales as well if inconsistencies are found.  

Author Response

Dear Reviewer 2
Thank you very much for this informative review shared along with respectful review points and vital questions that necessitate further improvement in many aspects of our article to cover each and every aspect of this vital subject.
Below are the answers to your respectful review points.

  1. There are some grammatical errors. For example, “We also assessment the best overall PSA response …” (Line 158), “Surgical interventions, including radical prostatectomy or metastectomy.” (Line 215, not a complete sentence), and “To date, there are a several clinical trials …” (Line 328).
  • We appreciate you pointing out these errors. These were all corrected and can be tracked in lines 160, 217, and 331 (gray highlights).

  1. Table 1: Initial ECOG Performance Status for 1 should be 43.4%, not 44.4%. Also, footnote: “3SD” should be “4SD” and there should be a “4” at “SD” of the “Mean ± SD”.
  • We appreciate you pointing out these typos. Amend and highlighted in gray.

  1. Section 3.4: “The median treatment activity for [177Lu]Lu-PSMA RLT was 22.2 GBq, with an IQR of 14.8 to 29.6 GBq. In contrast, the mean administered radioactivity for the [161Tb]Tb-PSMA treatments was 11.1 GBq, with a SD of ± 3.8 GBq.”. Why did the authors use two different expression methods (median and IQR vs mean and SD) for the dosages of [177Lu]Lu-PSMA and [161Tb]Tb-PSMA? Ideally, they should be the same, so it would be easier to compare their differences.
  • We appreciate your suggestion. Unfortunately, we cannot present [161Tb]Tb-PSMA radioactivity values the same [177Lu]Lu-PSMA values were expressed. This is simply because [161Tb]Tb-PSMA radioactivity variable was normally distributed while [177Lu]Lu-PSMA radioactivity values was not. We have used Shapiro-Wilk W test to test normality distribution as highlighted in line 186.

  1. Table 6: “18” for Any Grade and “16” for Grade 1. What are the grade for the other 2 patients?
  • Added in Table 7 highlighted in gray ( the grade was 2 )

  1. Lines 340-342: “Moreover, 3 months after the last injection, 40%, 37%, and 37% of patients had nonprogressive disease (complete response, partial response, and stable disease), respectively”. Double check the numbers as 40% + 37% + 37% is > 100%.
  • We appreciate you pointing out these typos. This was corrected in the updated version. Kindly track changes in lines 343-346 (gray highlights).

  1. The Discussion section is poorly written. This section is used to discuss the reported data and compare them with those reported by others. However, for the entire Discussion section, the authors mentioned only the data reported by others, and there were no comparisons at all with their own data. For example, in the 3rd paragraph, the authors mentioned the response rates reported by others. The authors should compare their response rates with those reported by others, and see if they are consistent. If not, the authors should provide rationales for the inconsistency. Similarly, for the 4th and 5th paragraphs, the authors mentioned the prediction factors reported by others. The authors should compare their data with those reported by others and see if they are consistent. The authors should provide rationales as well if inconsistencies are found.
  • Thank you for this impactful suggestion. The discussion section was reorganized. First, we have removed the fourth and fifth paragraph as it was mainly centered on discussing predictive factors which were not examined in our study. Second, we have added a new table (Table 8) to detail key similarities and difference between our study and selection of previous research works in response to your respectful review point (gray highlights). Third, we have detailed toxicity patterns reported in most recent meta-analysis in lines 384-393 (yellow highlights) to provide more insightful details.

Round 2

Reviewer 2 Report

Comments and Suggestions for Authors

This is a revised version and the authors have provided satisfactory responses and changes based on the reviewer's comments. Therefore, this revised manuscript could be accepted for publication without further changes.